# Magnetic field screening in hydrogen-rich high-temperature superconductors

V. S. Minkov [1✉], S. L. Bud'ko [2,3], F. F. Balakirev [4], V. B. Prakapenka[5], S. Chariton[5], R. J. Husband [6], H. P. Liermann [6] & M. I. Eremets [1✉]

In the last few years, the superconducting transition temperature, $T_c$, of hydrogen-rich compounds has increased dramatically, and is now approaching room temperature. However, the pressures at which these materials are stable exceed one million atmospheres and limit the number of available experimental studies. Superconductivity in hydrides has been primarily explored by electrical transport measurements, whereas magnetic properties, one of the most important characteristic of a superconductor, have not been satisfactory defined. Here, we develop SQUID magnetometry under extreme high-pressure conditions and report characteristic superconducting parameters for $Im$-$3m$-$H_3S$ and $Fm$-$3m$-$LaH_{10}$—the representative members of two families of high-temperature superconducting hydrides. We determine a lower critical field $H_{c1}$ of ~0.82 T and ~0.55 T, and a London penetration depth $\lambda_L$ of ~20 nm and ~30 nm in $H_3S$ and $LaH_{10}$, respectively. The small values of $\lambda_L$ indicate a high superfluid density in both hydrides. These compounds have the values of the Ginzburg-Landau parameter $\kappa$ ~12–20 and belong to the group of "moderate" type II superconductors, rather than being hard superconductors as would be intuitively expected from their high $T_c$s.

[1] Max Planck Institute for Chemistry, Hahn Meitner Weg 1, 55128 Mainz, Germany. [2] Ames Laboratory, U.S. Department of Energy, Iowa State University, Ames, IA 50011, USA. [3] Department of Physics and Astronomy, Iowa State University, Ames, IA 50011, USA. [4] Los Alamos National Laboratory, Los Alamos, NM 87545, USA. [5] Center for Advanced Radiation Sources, University of Chicago, 5640 South Ellis Avenue, Chicago, IL 60637, USA. [6] Photon Science, DESY, Notkestrasse 85, 22607 Hamburg, Germany. ✉email: v.minkov@mpic.de; m.eremets@mpic.de

The Bardeen–Cooper–Schrieffer[1] and Migdal–Eliashberg[2,3] theories of conventional phonon-mediated superconductivity imply that high frequency phonons and strong electron-phonon interactions are favorable for high-temperature superconductivity. Hydrogen, which has the highest naturally-occurring phonon frequencies due to its low mass, could be the best candidate material for high-temperature superconductivity[4,5]. Although the realization of superconductivity in pure hydrogen has been hindered by the extreme pressures required to reach the superconducting state (∼500 GPa), the idea of "chemical precompression" of hydrogen by heavier chemical elements in hydrogen-rich compounds[6] has brought great success. Following the discovery of $T_c = 203$ K in $H_3S$ at ∼150 GPa[7,8], higher $T_c$s were subsequently reported in so-called metal superhydrides including $T_c$ ∼220 K in $CaH_x$[9,10], $T_c$ ∼243 K in $YH_9$[11] and $T_c$ ∼250 K in $LaH_{10}$[12–14]. These major leaps toward room temperature superconductivity are the result of fruitful synergy between theory, computation, and experiment.

Superconductivity in hydrogen-rich compounds has since been demonstrated in numerous experiments[15,16]; however, it was identified based mostly on electrical transport measurements. Magnetic measurements, which are, inter alia, a crucial and independent test of superconductivity are scarce. They have not provided reliable experimental values of a lower critical field $H_{c1}$ and the London penetration depth $\lambda_L$ in hydrogen-rich superconductors. Magnetic field screening in the superconducting state of the $Im$-$3m$-$H_3S$ phase below 203 K was demonstrated using a superconducting quantum interference device (SQUID) and was in good agreement with the sharp drop of resistance in corroborating electrical resistance measurements of the same sample[7]. However, $H_{c1}$ was only roughly estimated from the hysteretic loops of $M(H)$ data instead of the initial virgin portion of magnetization curves of zero-field-cooled (ZFC) sample. More recently, the diamagnetic response in $H_3S$[17] and $LaH_{10}$[18] were qualitatively demonstrated by alternating current magnetic susceptibility measurements adapted for diamond anvil cells (DACs)[19].

In the present work, we created an effective approach for accurate magnetometry measurements of samples under megabar pressures by measuring the reference magnetic signal of the whole DAC assembly before the synthesis of a superconducting compound. This technique allows us to accurately determine the values of $H_{c1}$, $\lambda_L$, the Ginzburg–Landau parameter $\kappa$, and the critical current density $j_c$ in $Im$-$3m$-$H_3S$ and $Fm$-$3m$-$LaH_{10}$ high-temperature superconductors.

## Results

### Synthesis and characterization of superconducting samples.
Samples of $H_3S$ and $LaH_{10}$ were synthesized via a chemical reaction between sulfur or lanthanum trihydride and hydrogen at high pressures, in the stability field of the final products. The samples were prepared by sandwiching thin plates of S or $LaH_3$ between two thicker layers of $NH_3BH_3$ and pressurized in miniature nonmagnetic DACs to ∼170 GPa. The reference background magnetization signal was collected from the whole assembly of DAC including the pressurized precursor compounds in a SQUID magnetometer. The samples were subsequently heated using a pulsed laser to synthesize the desired superconducting products. Several photos of samples are shown in Fig. 1. Ammonia borane was chosen as an alternative source of hydrogen[20], as it readily decomposes at high temperature and releases free $H_2$. This approach was successfully implemented for synthesis of hydrides earlier[11,13]. In contrast to synthesis in an atmosphere of pure $H_2$, the use of $NH_3BH_3$ simplified the experimental procedure and significantly enlarged a size of the final products. The latter is crucial for SQUID measurements because the measured magnetic moment is proportional to square of a sample radius. In addition, the use of $NH_3BH_3$ allowed for correct reference magnetization measurements in contrast to pure hydrogen, which can cause uncontrolled spontaneous hydrogenation and formation of the superconducting phases at high pressure prior to the laser heating.

Figure 1g, h shows X-ray diffraction patterns from the dominant $Im$-$3m$-$H_3S$ and $Fm$-$3m$-$LaH_{10}$ phases in the heated samples. Although $LaH_{10}$ sample contained $P6_3/mmc$-$LaH_{10}$ as a minor impurity phase, we note that this was also found in various samples synthesized from La or $LaH_3$ and pure $H_2$ in previous studies and did not hinder superconductivity in $Fm$-$3m$-$LaH_{10}$, which has the highest $T_c$ in the lanthanum-hydrogen system[12,21]. Both the $Im$-$3m$-$H_3S$ and $Fm$-$3m$-$LaH_{10}$ phases were found to be homogeneous and evenly distributed within the heated area to an average diameter of ∼85 and ∼70 μm, respectively (insets in Fig. 1), which are in good agreement with the values estimated by optical microscopy. In addition, a sample pressure of $P_S = 155 \pm 5$ GPa for $Im$-$3m$-$H_3S$ and $P_S = 130 \pm 8$ GPa for $Fm$-$3m$-$LaH_{10}$ was determined more precisely based on the variation of the refined lattice parameters across the sample (see Supplementary Information). The $Fm$-$3m$ crystal lattice of $LaH_{10}$ sample is likely slightly distorted as (111), (220) and (311) diffraction peaks are broader than (200) peak. These peaks were shown to be most sensitive to the monoclinic structural distortions in $LaH_{10}$ at pressures below ∼138 GPa[21].

### $M(T)$ magnetization measurements.
The ZFC samples with the $Im$-$3m$-$H_3S$ and $Fm$-$3m$-$LaH_{10}$ phases exhibit clear diamagnetic signal below their respective $T_c$s, indicated that they had become superconducting after laser heating (see Fig. 2 and Supplementary Figs. S1 and S2). The pronounced changes were detected in raw voltage curves of direct current (DC) scans measured before and after laser heating (see Fig. 2a, b). The pressurized unheated precursors have two minima in DC scans at ∼0 and −1.5 cm, which correspond to the centered position of the sample in DAC and the center of the massive part of DAC body including a piston and a cap, respectively (see Supplementary Fig. S3). After the synthesis of $Im$-$3m$-$H_3S$ and $Fm$-$3m$-$LaH_{10}$ phases, an additional diamagnetic signal originating from the superconductor below its $T_c$ appeared at 0 cm. The distinctive step on the resulting $M(T)$ dependence associated with superconductivity was observed in both heated samples at 2, 4 and 10 mT (see Fig. 2c–f). The $LaH_{10}$ sample has a broader superconducting transition, which is most likely caused by a larger pressure gradient across the sample and a strong $T_c(P)$ dependence on the verge of structural instability in this pressure range[21]. The observed values of $T_c$ ∼231 K in $Fm$-$3m$-$LaH_{10}$ and ∼196 K in $Im$-$3m$-$H_3S$ are in excellent agreement with the previously-reported values from four-probe electrical transport measurements of samples at the same pressures[7,21–24].

It is worth noting that whereas the superconducting transition is pronounced in ZFC measurements, its signature is subtle or almost undetectable in field-cooled measurements (see Fig. 2g, h and Supplementary Figs. S1 and S2). The weak flux expulsion or its absence is well-known for type II superconductors with strong pinning of vortices[25]. Strong pinning prevents vortices inside the sample from leaving the sample below the $H_{c1}(T)$ value. The very low fields are favorable for the detection of the Meissner state, because in this case the $H_{c1}(T)$ line is crossed in the vicinity of $T_c$ where critical currents are smaller and the pinning is weaker[26]. We also observed strong suppression of the Meissner effect in the test measurements performed on a powder sample of $MgB_2$ (see Supplementary Information). No flux expulsion at all was also reported in some publications on Fe-based superconductors[27,28].

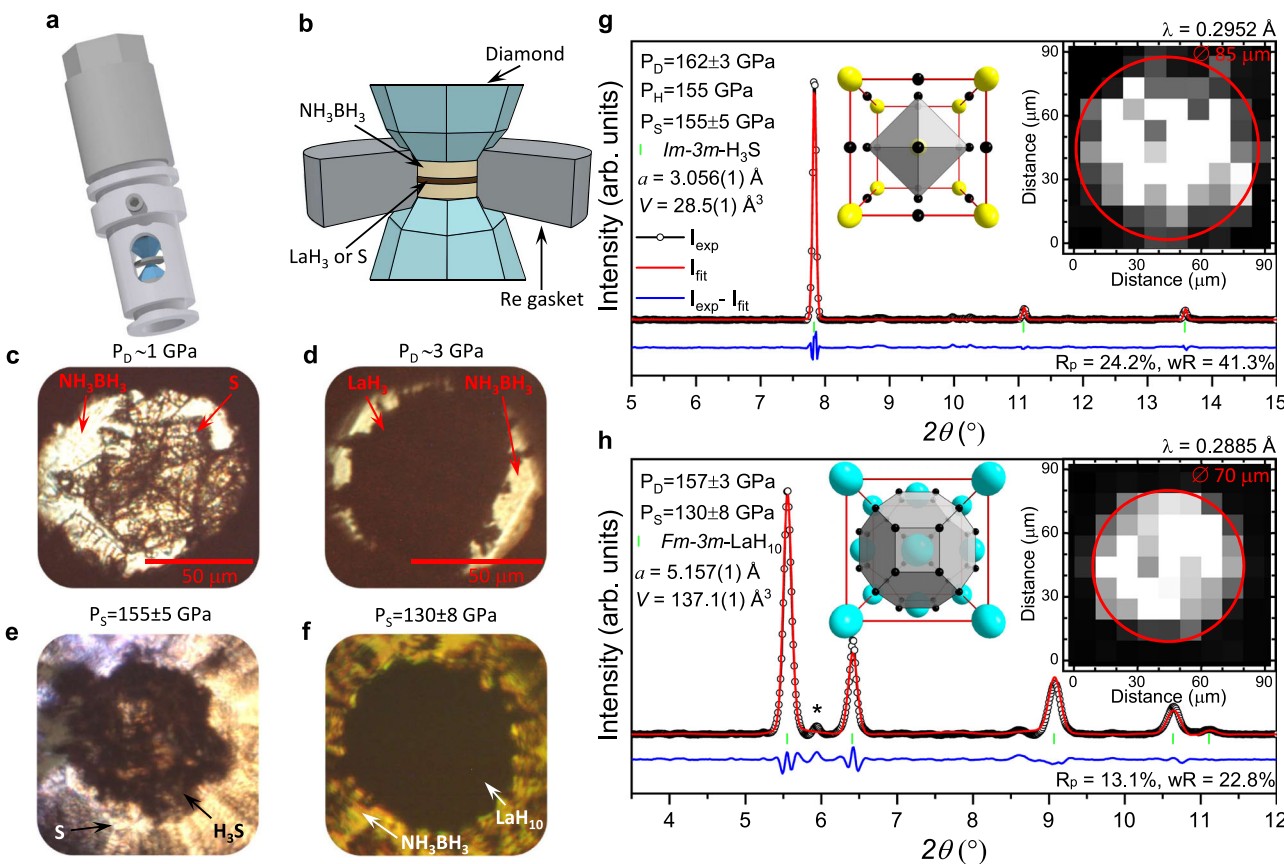

**Fig. 1 Synthesis of the _Im-3m_-H$_3$S and _Fm-3m_-LaH$_{10}$ phases. a** Illustration of the miniature DAC used for magnetic measurements in SQUID. **b** Scheme showing the typical arrangement of the sandwiched precursors in the DAC. **c, d** Photos of the sandwiched samples, S + NH$_3$BH$_3$ and LaH$_3$ + NH$_3$BH$_3$, after loading at $P_D$ ~1 and ~3 GPa, respectively. **e, f** Photos of the H$_3$S and LaH$_{10}$ samples after compression and subsequent pulsed laser heating. **g, h** X-ray powder diffraction patterns collected from the synthesized _Im-3m_-H$_3$S and _Fm-3m_-LaH$_{10}$ samples. The black circles and red and blue curves correspond to the experimental data, Rietveld refinement fits and residues, respectively. The green ticks indicate the calculated peak positions. The (101) reflection stemming from the _P6$_3$/mmc_-LaH$_{10}$ impurity phase is marked by asterisk. The fragments of the crystal structure with the characteristic SH$_6$ and LaH$_{32}$ coordination polyhedra are shown as insets. The large yellow and cyan and small black spheres represent the S, La and H atoms in the crystallographic unit cells, respectively. The spatial distribution across the heated samples and the estimated diameter of the superconducting _Im-3m_-H$_3$S and _Fm-3m_-LaH$_{10}$ phases are shown as inset. $P_D$, $P_H$ and $P_S$ are pressure values estimated from the position of diamond edge and hydrogen vibron in Raman spectra and refined lattice parameters of the final products, respectively (see details in "Methods").

When measuring a superconducting sample with a large magnetization value, it is important to consider demagnetization effects. The total magnetic field $H_t$ inside a sample is given by:

$$H_t = H - H_d, \quad (1)$$

where $H$ is the external applied field and $H_d$ is the demagnetization field. The demagnetization field is given by $H_d = -NM$, where $N$ is a shape-dependent demagnetization factor and $M$ is the magnetization of a sample. For a long and thin sample in a parallel field $N \approx 0$, while for a short and flat sample in a perpendicular magnetic field the demagnetization correction $NM$ can be enormous.

We estimated an effective demagnetization factor $N$ in the studied samples using the measured values of magnetization, assuming an ideal diamagnetic signal in low fields ZFC measurements. Then the demagnetization correction for a perfect diamagnet (magnetic susceptibility $\chi = -1$) can be written as follows:

$$\frac{\triangle M}{HV} = -\frac{1}{1-N}, \quad (2)$$

where $V$ is the volume of a sample. The absolute value of $\Delta M$, the difference in $M$ between a normal metal state (above $T_c$) and a superconducting state (below $T_c$), was extracted from the

measurements by subtraction of the reference data of the compressed sandwiched samples (see Fig. 2). A thickness of samples is, however, the main contributor to the uncertainty in the values of $V$ and $N$ because it cannot be directly probed in the experiment. Nevertheless, we indirectly estimated a thickness of the final H$_3$S and LaH$_{10}$ samples as ~2.8 and ~1.9 μm, respectively, by considering: (1) the visual expansion of samples during pressurizing, (2) the pressure-induced compressibility of S and LaH$_3$, and (3) the increase of product volumes after the hydrogenation reaction (see details in Supplementary Information). If we put the values of $\Delta M$ and $V$ in Eq. (2) the demagnetization correction is ~8.5 for the sample of _Im-3m_-H$_3$S and ~13.5 for the sample of _Fm-3m_-LaH$_{10}$.

An alternative way of the evaluation of the diamagnetic factor would be approximating of the shape of samples by thin solid disks and using the equation for effective demagnetization factor from ref. [29] (see detailed discussion in "Methods"). We consider the $N$ values from $M(T)$ data more reasonable and reliable and will use them in the rest of the paper.

**$M(H)$ magnetization measurements.** Measurements of the magnetic field dependence of magnetization allow us to estimate the characteristic superconducting parameters $H_{c1}$, $\lambda_L$, $\kappa$, and $j_c$.

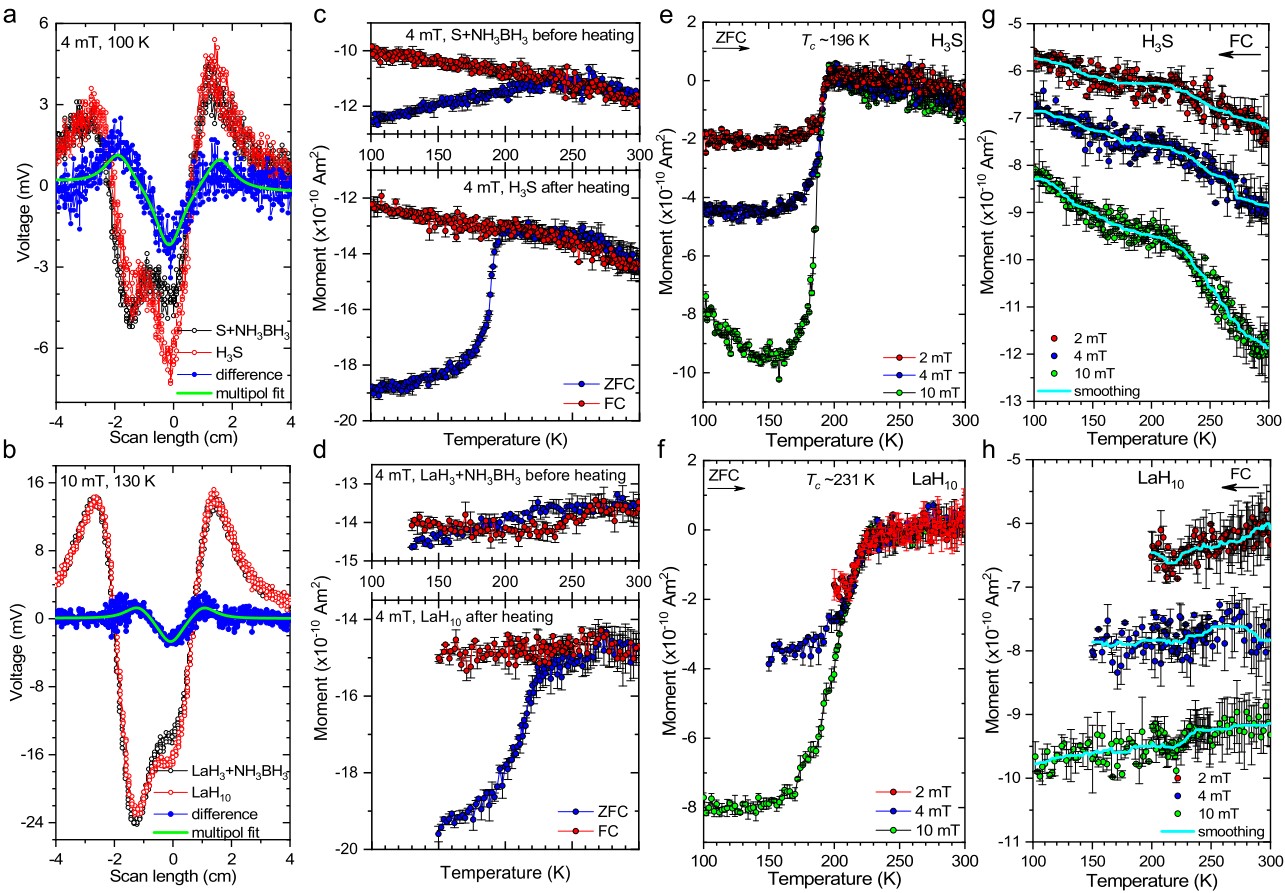

**Fig. 2 Expulsion of magnetic field by the superconducting *Im-3m*-H₃S and *Fm-3m*-LaH₁₀ phases at $P_S = 155 ± 5$ GPa and $P_S = 130 ± 8$ GPa, respectively.**
**a**, **b** Appearance of the diamagnetic signal in the raw SQUID output signal of ZFC samples containing the superconducting *Im-3m*-H₃S and *Fm-3m*-LaH₁₀ phases under low magnetic fields below their $T_c$s. Black circles correspond to data measured for the compressed S + NH₃BH₃ and LaH₃ + NH₃BH₃ sandwiched samples at $P_D \sim 167$ GPa before laser heating, red circles correspond to data collected from heated samples with *Im-3m*-H₃S and *Fm-3m*-LaH₁₀ phases. The blue circles show the difference, whereas the green curve shows the fit. The position at 0 cm corresponds to the center of the pickup coil relative to which the superconducting samples were centered (see "Methods"). **c**, **d** ZFC and FC portions of $M(T)$ measurements of the sandwiched samples with metallic S and LaH₃ (top panel) and heated samples with *Im-3m*-H₃S and *Fm-3m*-LaH₁₀ phases (bottom panel), respectively. **e**, **f** ZFC portions of $M(T)$ magnetization data for the *Im-3m*-H₃S and *Fm-3m*-LaH₁₀ phases at 2, 4 and 10 mT after subtraction of the background signal measured from the DACs with the pressurized unheated sandwiched samples. **g**, **h** FC portions of $M(T)$ data measured at 2, 4 and 10 mT after formation of the *Im-3m*-H₃S and *Fm-3m*-LaH₁₀ phases. The curves were vertically translated for better representation. The raw $M(T)$ data of the initial pressurized and heated samples are summarized in Supplementary Figs. S2 and S3. Smoothed by a percentile filter, light blue curves demonstrate the subtle Meissner effect in FC measurements.

The value of $H_p$, at which the applied magnetic field starts to penetrate the sample, was determined from the onset of the deviation of $M(H)$ from the linear dependence (see Fig. 3). The extrapolation of $H_p(T)$ to lower temperatures yields $H_P(0\,K)$ ~96 mT for *Im-3m*-H₃S and ~41 mT for *Fm-3m*-LaH₁₀. Applying the demagnetization correction we obtain values of $H_{c1}(0\,K)$ ~820 mT for H₃S and ~550 mT for LaH₁₀. The Ginzburg–Landau parameter $\kappa$ can be evaluated from the equation:

$$\frac{H_{c2}}{H_{c1}} = \frac{2\kappa^2}{\ln\kappa}, \quad (3)$$

where $H_{c2}$ is an upper critical field[30]. Inserting the experimental estimations of $H_{c2}(0\,K)$ ~97 T for *Im-3m*-H₃S[23] and ~143.5 T for *Fm-3m*-LaH₁₀[21] gives $\kappa$ ~12 and ~20 for H₃S and LaH₁₀, respectively. A coherence length $\xi(0\,K)$ ~1.8 nm for H₃S and ~1.5 nm for LaH₁₀ were evaluated using the available experimental data[21,23], which gives a London penetration depth of $\lambda_L(0\,K)$ ~22 nm in *Im-3m*-H₃S and ~30 nm in *Fm-3m*-LaH₁₀. The temperature dependence of $\lambda_L$ is shown in Supplementary

Fig. S4. At low temperatures, the *s*-wave model of conventional superconductivity well reproduces the data for both compounds. The thermodynamic critical field value is given by:

$$H_c(0\,K) = \frac{\sqrt{H_{c1}(0\,K)H_{c2}(0\,K)}}{\sqrt{\ln\kappa}}. \quad (4)$$

$H_c(0\,K)$ ~5.6 T for H₃S and ~5.1 T for LaH₁₀. The robustness of the dissipation-free vortex solid phase in hydrides can be evaluated via Ginzburg–Levanyuk number:

$$Gi = \frac{1}{2}\left(\frac{2\pi\mu_0\kappa_B T_c\lambda_L^2}{\phi_0^2\xi}\right)^2 \quad (5)$$

were $\kappa_B$ is the Boltzmann constant, $\mu_0$ is the vacuum permeability and $\phi_0$ is the magnetic flux quantum. $Gi$ quantifies the scale of fluctuations responsible for vortex melting[31] and vortex creep[32] in a superconductor. $Gi(0\,K)$ ~9 × 10⁻⁷ for H₃S and ~6 × 10⁻⁶ for LaH₁₀, which is substantially smaller than what is reported for cuprate and pnictide high-temperature superconductors[31,32] and comparable to that of Nb₃Sn. Despite high $T_c$ both hydrides display a moderate $\kappa$ which results in weaker vortex fluctuations

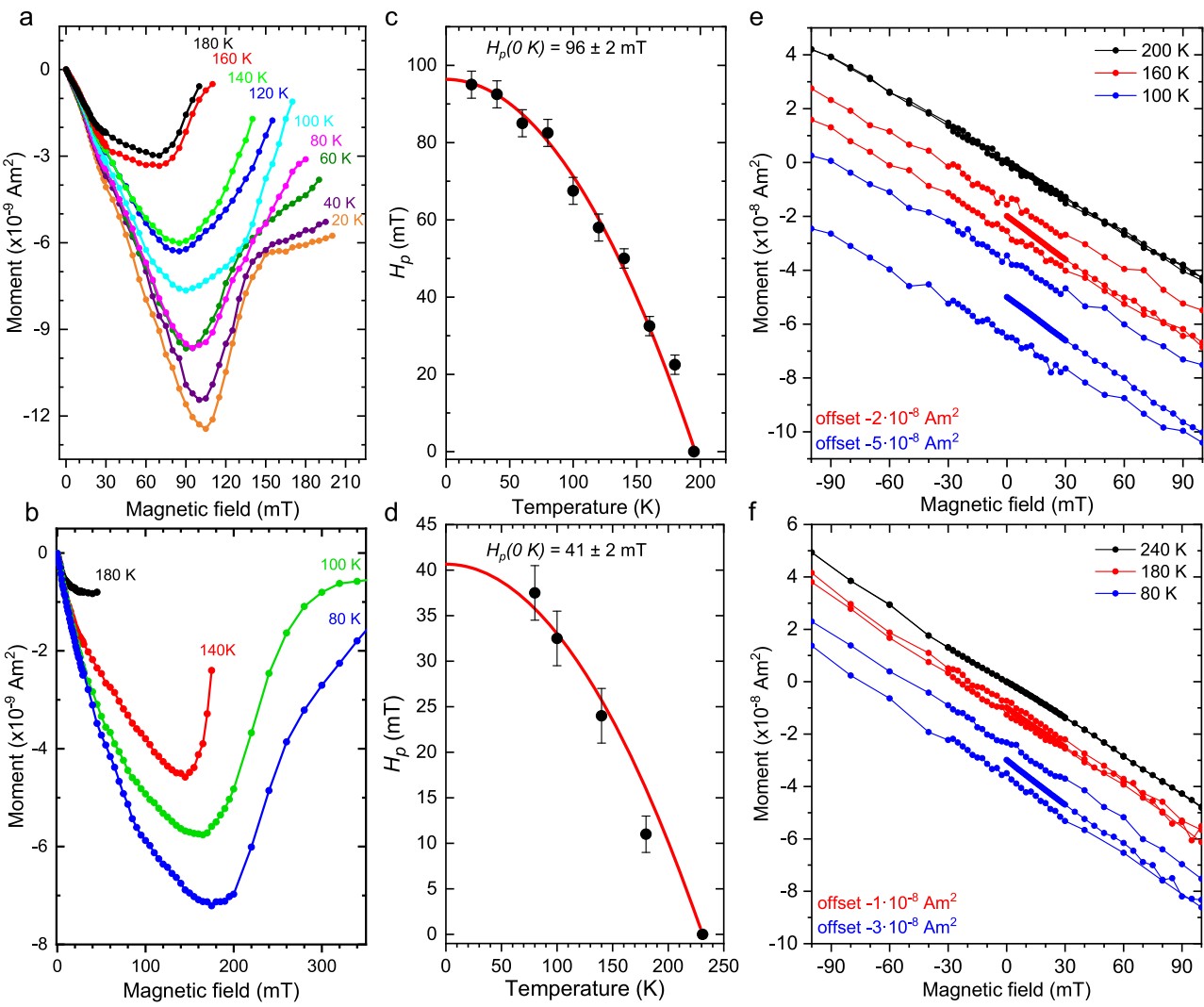

**Fig. 3 $M(H)$ magnetization data for $Im\text{-}3m$-$H_3S$ and $Fm\text{-}3m$-$LaH_{10}$ at high pressure. a, b** Magnetic moment associated with the penetration of the applied magnetic field into the $Im\text{-}3m$-$H_3S$ phase at $P_S = 155 \pm 5$ GPa and the $Fm\text{-}3m$-$LaH_{10}$ phase at $P_S = 130 \pm 8$ GPa based on virgin curves of the $M(H)$ magnetization data at selected temperatures. The curves were superimposed by performing linear transformations for a better representation. A linear background, defined as a straight line connecting $M(H = 0 \text{ T})$ and $M(H = 1 \text{ T})$ at corresponding temperature, was subtracted. After that the data were normalized to $H = 15$ mT data so that to have the same initial linear $M(H)$ slope. **c, d** Temperature dependence of a penetration field $H_p$ for $Im\text{-}3m$-$H_3S$ and $Fm\text{-}3m$-$LaH_{10}$ derived from the virgin curves of $M(H)$ magnetization data. Black circles and red curves correspond to the experimental data and fits, respectively. **e, f** Hysteretic part of $M(H)$ measurements of $H_3S$ and $LaH_{10}$ in the normal metallic state above $T_c$ and the superconducting state below $T_c$, correspondingly.

and explains why the reported vortex liquid region remains narrow even at high magnetic fields[21,23].

It is important to estimate the range of values of the superconducting parameters of $Im\text{-}3m$-$H_3S$ and $Fm\text{-}3m$-$LaH_{10}$, which depends on a sample thickness. We evaluated the minimum and maximum values of a sample thickness as 2.1–3.1 μm for $H_3S$ and 0.6–2.5 μm for $LaH_{10}$ (see details in Supplementary Information). Thus, the lower and upper limits are: ~0.74–1.09 T for $H_{c1}(0 \text{ K})$, ~18–23 nm for $\lambda_L(0 \text{ K})$ and ~10–13 for $\kappa$ in $Im\text{-}3m$-$H_3S$; and ~0.42–1.75 T for $H_{c1}(0 \text{ K})$, ~14–35 nm for $\lambda_L(0 \text{ K})$ and ~10–23 for $\kappa$ in $Fm\text{-}3m$-$LaH_{10}$. For $LaH_{10}$, the larger dispersion is a result of a substantial increase of $N$ in case of the lower limit of a sample thickness. The estimated superconducting parameters for both hydrides are summarized in Table 1.

A general behavior of $M(H)$ (see Fig. 3e, f) is typical for type II superconductors, in particular, the difference in a magnetic moment between the forward and reverse sweep of an applied

magnetic field increases with a decrease of temperature. However, it was not possible to observe the extent of the superconducting magnetization hysteresis and the characteristic field $H^*$, above which magnetization becomes reversable, because the magnetic signal of the DAC becomes much higher in comparison with the magnetic response of $H_3S$ and $LaH_{10}$ samples for the reliable subtraction of the background.

We also estimated the critical currents using the Bean critical state model[33,34]. This model assumes that when the sweeping magnetic field fully penetrates the sample the density of the screening current equals the critical current value $j_c$. The difference in magnetic moment $\Delta m$ between forward and reverse field is due to the reversal of the direction of the screening moment, and can be used to evaluate the magnitude of the bulk critical current. A concentric screening current pattern with density $j_c$ creates a magnetic moment $m = \frac{\pi}{3} j_c h r^3$, where $h$ is the thickness and $r$ is the radius of the disk-shaped sample, often approximated by $j_c = 30 \frac{m}{Vr}$, where $m$, $V$, and $r$ are in CGS units

**Table 1 Summary of estimated superconducting parameters for *Im-3m*-H₃S and *Fm-3m*-LaH₁₀ at high pressure.**

| Sample | $P_S$, GPa | $T_{cr}$ K | Size, μm | | $\frac{1}{1-N}$ | $H_p$(0 K), mT | $H_{c1}$(0 K), T | $\lambda_L$, nm | $\kappa$ | $Gi$ |
|---|---|---|---|---|---|---|---|---|---|---|
| | | | ∅ | Thickness | | | | | | |
| *Im-3m*-H₃S | 155 ± 5 | ~196 | 85 | 2.8 (2.1–3.1) | 8.5 (7.7–11.4) | 96 ± 2 | 0.82 (0.74–1.09) | 22 (18–23) | 12 (10–13) | $9 \times 10^{-7}$ |
| *Fm-3m*-LaH₁₀ | 130 ± 8 | ~231 | 70 | 1.9 (0.6–2.5) | 13.5 (10.2–42.6) | 41 ± 2 | 0.55 (0.42–1.75) | 30 (14–35) | 20 (10–23) | $6 \times 10^{-6}$ |

The upper and lower limits of corresponding parameters are parenthesized.

and $j_c$ is in A cm$^{-2}$. We found that $j_c$ reaches values of ~$7 \times 10^6$ A cm$^{-2}$ for both LaH₁₀ and H₃S at 100 K (see Supplementary Fig. S5). This high value of the critical current indicates strong vortex pinning, which corroborates negligible magnetization signal observed during field cooling, as well as very high irreversibility field $H^*$, or vortex melting/de-pining field, reported in magnetotransport measurements[7,12,21,23,35]. The critical currents in LaH₁₀ estimated from the magnetization measurements are of the same order of magnitude but somewhat higher than the value of $1.2–2.8 \times 10^6$ A cm$^{-2}$ at 4.2 K measured in presumably yttrium-doped lanthanum superhydride using electrical transport technique[35]. This discrepancy can be attributed to the fact that in the transport measurements $j_c$ is constrained by those parts of the electrical current path where the superconductivity is least robust, while the magnetization signal is dominated by the parts of the sample with most robust superconductivity.

## Discussion

It is informative to compare our findings with the available magnetic data from previous studies on H₃S[7,36]. Although $H_{c1}$ ~30 mT was reported in an earlier study of *Im-3m*-H₃S[7], this value was strongly underestimated. Firstly, $H_{c1}$ was determined from the hysteretic loops of $M(H)$ data instead of the initial virgin magnetization curves, which were not measured. Secondly, the real shape of the superconducting phase was not determined and the corresponding demagnetization correction were not applied. In addition, the diamagnetic signal was superposed by a much stronger paramagnetic signal, presumably stemming from the body of DAC.

In another work, authors applied forward nuclear resonant scattering technique using the $^{119}$Sn Mössbauer isotope as a sensor and reported a value of $H$ ~0.68 T, which was expelled by the sample at ~120 K[36]. The sample was synthesized by pressure-induced disproportionation of H₂S at ~150 GPa as in ref. [7], however the superconducting phase was not characterized—neither crystal structure nor $T_c$ were determined. Since the geometry and arrangement of a tin foil and superconducting sample, which are required for calculations of demagnetization and end effects, are unknown, it is not possible to quantify the value of $H_{c1}$ from this experiment.

In summary, we have performed magnetization measurements using a specially-designed miniature DAC for representative members of two families of hydrogen-rich superconductors—H₃S, which contains covalent H-S bonds, and LaH₁₀, which has ionic bonding between La and H. The present data demonstrate that the diamagnetic signal is absent in the pressurized S and LaH₃ precursor compounds and only appears after laser heating of the samples and the resultant chemical synthesis of the respective superconducting phases. In contrast to high-$T_c$ superconductors of the cuprate family, the *Im-3m*-H₃S and *Fm-3m*-LaH₁₀ phases have significantly lower values of $\lambda_L$. The low values of $\kappa$ indicate that both compounds belong to "moderate" type II superconductors not far from the clean limit. Both H₃S and LaH₁₀ hydrides possess good superconducting characteristics; in addition to high values of $T_c$ they exhibit high critical current

densities and have high values of lower and upper critical fields. These make hydrogen-rich compounds promising materials for technological use, provided that they can be stabilized at ambient or accessible pressure conditions.

## Methods

**Diamond anvil cell**. The samples were synthesized in miniature DACs, which were specially designed for a standard commercial SQUID magnetometer (either Quantum Design MPMS or Cryogenic Limited S700X) with the sample space diameter of 9 mm by reworking and modifying the prototype piston/locking nut design. The design of the DAC was briefly described in ref. [7]. To minimize the magnetic signal over a wide temperature range simultaneously providing a high mechanical strength, the body of the DAC was made of a high-purity Cu-Ti alloy with 3 wt% Ti[37]. This material has the lowest magnetic susceptibility among the known hard metallic alloys: $\chi_g = 8 \times 10^{-4}$ mJ T$^{-1}$ g$^{-1}$ at 1.8 K and at the same time it is hard enough to build parts of the DAC: its tensile strength is ~$10^3$ MPa[38]. As parts such as piston and diamond seats are subjected to the highest load, they were made of harder Cu-Be alloy with 1.8-2.0 wt% Be. Such combination of materials allows us to construct the miniature DAC with an outer diameter of 8.8 mm, which is capable to reach pressures as high as 220 GPa retaining the low overall magnetic response.

The diamonds were beveled at 9° to a diameter of ~250 μm with a culet size of ~75 and ~90 μm. In total, 200-μm-thick rhenium gasket was pre-indented to a thickness of 20 and 30 μm, and a hole with a diameter of about the culet size was drilled using a laser. All elements of the high-pressure cell assembly were thoroughly etched in acids in order to remove a possible contamination by magnetic pieces, which could stem from the manufacturing of the DAC parts, polishing of the diamonds and cutting of the gaskets. All parts of the DACs and prepared gaskets were etched in 3 M hydrochloric acid for 30 min, and diamonds were etched in a mixture of concentrated nitric and hydrochloric acids in 1:3 molar ratio for 90 min in an ultrasonic cleaner.

**Preparation of samples**. Sulfur (99.999%, Alfa), NH₃BH₃ (97%, Sigma-Aldrich), and LaH₃, which was synthesized from La (99.9%, Alfa Aesar) and H₂ (99.999%, Spectra Gases), were used as initial reactants. In contrast to metallic La, LaH₃ was beneficial because it required less hydrogen for the full hydrogenation. The loading of samples in DACs were handled in an inert Ar atmosphere with the O₂ and H₂O residual contents of <0.1 ppm. NH₃BH₃ acted both as a source of H₂ and a thermal isolator from the diamonds during laser heating. The thin plates of S, LaH₃ and NH₃BH₃ for the sandwiched samples were molded out the corresponding powder samples by squeezing them between two large 1-mm-diameter diamond anvils. The thickness of the plates was monitored by the interference of the visible light.

The sandwiched samples, in which 8-μm-thick S or 6-μm-thick LaH₃ plates were interposed between two ~10–15-μm-thick layers of NH₃BH₃ were put in the hole of pre-indented metallic gaskets. Then samples were pressurized to $P_D$ of ~167 GPa with a pressure gradient across the culet of about ±7 GPa. The decomposition of NH₃BH₃ and synthesis of the superconducting *Im-3m*-H₃S and *Fm-3m*-LaH₁₀ phases were performed using the one-side heating with Nd:YAG pulse laser (a wavelength $\lambda = 1.064$ μm, the duration of pulses of 3 μs, and frequency of $10^4$ Hz). We heated S + NH₃BH₃ sample at ~700 K and LaH₃ + NH₃BH₃ sample at ~2000 K by traversing the ~5-μm-diameter laser spot horizontally and vertically across the diamond culets. Several photos of the initial, pressurized and heated samples are summarized in Supplementary Figs. S6 and S7.

Importantly, the integrity of a superconducting phase in a final sample is crucial for detecting of the diamagnetic signal by a SQUID. For example, LaH₃ completely transformed to the *Fm-3m*-LaH₁₀ phase already after the first laser heating at ~1000 K according to the X-ray diffraction data, nevertheless the superconducting transition was not observed in the magnetic measurements (see Supplementary Fig. S8). We guessed that this was because the sample was not uniform and consisted of separate parts, from which the sum magnetic signal is smaller than that from the one uniform disk of the same integral area. Our rough estimations gave a factor of ~5 difference in the signal between one 60-μm-diameter disk and 20 12-μm-diameter disks with a thickness of 2 μm just because of different demagnetization factors. Additionally, the smaller disks might have a smaller total volume resulting in an increase of a factor to ~10, so the sum magnetic signal becomes less than the sensitivity of a SQUID. To improve the integrity of the

superconducting phase by sintering, we again heated $LaH_{10}$ but at significantly higher temperatures of ~2000 K. As a result, the pronounced superconducting transition appeared in the subsequent magnetic measurements.

**Estimation of pressure**. The pressure values in samples were estimated using three different techniques. Initially we determined the pressure in the compressed sandwiched samples using the diamond scale[39] based on the shift of the Raman line edge of a stressed diamond (marked as $P_D$ in the text). This scale is not accurate and depends on the arrangement of a sample and the geometry of diamond anvils. Therefore, after the high-pressure synthesis we estimated pressure values more accurately using the refined lattice parameters of the superconducting $Im$-$3m$-$H_3S$ and $Fm$-$3m$-$LaH_{10}$ phases from X-ray diffraction data (marked as $P_S$ in the text). The average value of the refined lattice parameter $a$ across the sample is 3.057(8) Å for $Im$-$3m$-$H_3S$ phase and 5.175(13) Å for $Fm$-$3m$-$LaH_{10}$ phase. Taking into account the available accurate structural data of $H_3S$ and $LaH_{10}$ measured in samples under quasi hydrostatic conditions of $H_2$ medium[12,21–24,40], we defined pressure as $P_S$ = 155 ± 5 GPa in $Im$-$3m$-$H_3S$ sample and $P_S$ = 130 ± 8 GPa in $Fm$-$3m$-$LaH_{10}$ sample.

In addition, we identified the vibron of $H_2$ at ~4035 $cm^{-1}$ in Raman spectra of the heated sample with $Im$-$3m$-$H_3S$ phase (see Supplementary Fig. S8), which corresponds to $P_H$ = 155 GPa according to the hydrogen scale[41].

**X-ray diffraction measurements**. X-ray diffraction data were collected from the heated samples in the same miniature DACs at the beamlines 13-IDD at GSE-CARS, Advanced Photon Source ($\lambda$ = 0.2952 Å, a beam spot size of ~2.5 × 3.5 $\mu m^2$, Pilatus 1 M CdTe detector) and P02.2 at PETRA III, DESY ($\lambda$ = 0.2885 Å, a beam spot size of ~2 × 2 $\mu m^2$, LAMBDA GaAs detector). The reference samples of $LaB_6$ and $CeO_2$ were used for calibration of the distance between sample and detector. To examine the size and distribution of the superconducting phase in the samples, we collected X-ray powder diffraction patterns form the spatial area of 110 × 110 $\mu m^2$ with the horizontal and vertical step of 10 $\mu$m. Primary processing and integration of the data were made using the Dioptas software[42]. The indexing of X-ray diffraction patterns and refinement of the crystal structures were done with GSAS and EXPGUI packages[43].

**Magnetization measurements**. Magnetization measurements were done in the S700X SQUID magnetometer by Cryogenic Limited, a miniature DAC was attached to the 140-mm-long straw made of kapton polyimide film, which was specially designed to minimize the end effects. The relative position of the sample in the SQUID magnetometer was determined using the ferromagnetic signal from a small steel piece with a size of about 140 × 100 × 25 $\mu m^3$ attached directly to the rhenium gasket, which surrounded the sample (see Supplementary Fig. S3). This approach allows one to directly find the sample position in contrast to the centering procedure for a symmetric DAC, in which the total magnetization response from the whole assembly is inferred (presumed to be) as symmetric[44]. To minimize the errors associated with the sample positioning at different temperatures, the temperature-induced expansion of the rod, which holds the sample, was additionally calibrated within the wide temperature range using the ferromagnetic signal from the same steel piece.

For the micrometer-size superconducting samples in DACs we were able to extract the small diamagnetic signal of a superconductor from the measured overall magnetic moment including that of the bulky body of DAC, diamonds and rhenium gasket. We first measured the magnetic signal of the DACs with the starting pressurized precursor compounds before laser heating, in which S and $LaH_3$ were normal metals. Then we subtracted these reference data from the magnetic moment collected from the same DACs after laser heating and chemical synthesis, i.e., with the superconducting $Im$-$3m$-$H_3S$ and $Fm$-$3m$-$LaH_{10}$ phases.

$T_c$ was determined as the offset of the diamagnetic transition on the ZFC curves of the temperature dependence of magnetic moment $M(T)$. The other basic characteristic of superconductivity, such as $H_{c1}(0\,K)$, $\lambda_L(0\,K)$ and $\kappa$, were determined from the magnetization measurements. The $M(H)$ data were collected at several temperatures above and below $T_c$ within the range of magnetic field up to −1–1 T and summarized in Supplementary Figs. S10 and S11. The value of $H_p$, at which an applied magnetic field starts to penetrate into the sample, was determined from the onset of the evident deviation of the $M(H)$ from the linear dependence. We note that the raw magnetization curves include the significant diamagnetic response of the miniature high-pressure cell, which increases with the applied magnetic field (see Fig. 3e, f and Supplementary Figs. S10 and S11). To better illustrate the determination of $H_p$ in Figs. 3a and 3b, we have subtracted a linear background from the measured $M(H)$ magnetization data. This linear background was determined as the straight line connecting two endpoints: the magnetic moment value at $H$ = 0 T (the starting point of measurements) and the magnetic moment value at $H$ = 1 T (the highest value of the applied magnetic field) (see Supplementary Fig. S12). Subsequently, we performed additional linear transformations so that the curves have the same initial linear $M(H)$ slope. Importantly, these linear manipulations do not affect the onset of the deviation of the $M(H)$ virgin curve from the linear dependence. We extrapolated $H_p(T)$ to lower temperatures using the equation $H_p(T) = H_p(0\,K)(1 - (\frac{T}{T_c})^2)$ with the fixed $T_c$ = 196 K for $Im$-$3m$-$H_3S$ and 231 K for $Fm$-$3m$-$LaH_{10}$. The coherence length $\xi(0\,K)$ was determined from the experimental estimation of $H_{c2}(0\,K)$[21,23] using the

equation $H_{c2}(0\,K) = \frac{\phi_0}{2\pi\xi^2(0\,K)}$. The London penetration depth was determined using the equation $\lambda_L = \kappa\xi$.

Another, independent, very rough estimation of the $\lambda_L$ can be done using the equation $\lambda_L = \sqrt{\frac{m_e c^2}{4\pi n_s e^2}}$, where $m_e$ is the effective mass, $c$—speed of light, $n_s$—density of superconducting electrons, $e$—electron charge. Using $n$ for $n_s$ from the measurements of the Hall effect at room temperature[21,23] and assuming no mass enhancement the $\lambda_L(0\,K)$ is ~18 nm for $H_3S$ and ~63 nm for $LaH_{10}$. These values are reasonably consistent with the more accurate estimations using the values of the upper and lower critical fields.

An order-of-magnitude estimation of the mean free path $l$ can be done using the formula $l = \frac{m v_F}{ne^2 \rho(0\,K)}$, where $v_F = \frac{\xi(0\,K)\pi\Delta(0\,K)}{\hbar}$ is the Fermi velocity with $2\Delta(0\,K) = 3.52\,k_B T_c$, $\rho(0\,K)$—a residual resistivity, and $\Delta(0\,K)$—a superconducting gap. Using the data from refs. [21,23] we obtain the values for $v_F$ of ~2.6 × $10^5$ m $s^{-1}$ and $l$ of ~1.3 nm for $Im$-$3m$-$H_3S$, and $v_F$ of ~2.5 × $10^5$ m $s^{-1}$ and $l$ of ~4 nm or ~0.4 nm for $Fm$-$3m$-$LaH_{10}$ if we assume $\rho(0\,K) = 0.1 \times \rho(300\,K)$ or $\rho(0\,K) = \rho(300\,K)$, respectively. Thus, we conclude that $l \sim \xi(0\,K)$ for both materials and they are not far from the clean limit.

**Demagnetization correction**. We estimated the effective demagnetization correction as ~8.5 for the $\varnothing 85 \times 2.8\,\mu m^3$ sample of $Im$-$3m$-$H_3S$ and ~13.5 for the ~$\varnothing 70 \times 1.9\,\mu m^3$ sample of $Fm$-$3m$-$LaH_{10}$ using the measured value of magnetization. The demagnetization factor $N$ can be also calculated for the given geometry of a superconductor according to ref. [29]. This gives demagnetization correction ~20 and ~24 for the $Im$-$3m$-$H_3S$ and $Fm$-$3m$-$LaH_{10}$ samples, respectively. We note that the difference in values of $N$ for a thin disk- and thin ellipsoid-shaped (due to the cupping effect in diamond anvils at high pressures) samples is negligible if a diameter is much larger than a thickness. For the sake of simplicity, we consider the samples as thin disks in our estimates. The larger values of the computed demagnetization correction stem from the ignoring of variation of thickness in a sample and imperfections of sample integrity, which are, conversely, already included in the experimental value of $\Delta M$. A magnetic field penetrates into a superconductor at these imperfections decreasing an effective demagnetizing factor $N$. Thus, using the measured value of $\Delta M$ we obtain a lower estimate of $N$, since a decrease of a sample volume in a case of variation of a sample thickness will lead to an increase of $N$. For a good single-crystal, where imperfections of a sample are minimal, the demagnetization correction should be of the same value if calculated from a sample geometry or its value of magnetization. For instance, we have almost the same values of demagnetization correction derived from the prior ZFC $M(T)$ measurements of the test single-crystal of $Bi_2Sr_2CaCu_2O_8$ with a size of a size of $100 \times 80 \times 10\,\mu m^3$ (see details in Supplementary Information). In particular, $\frac{1}{1-N}$ ~8 from the value of $\Delta M$ and ~7 from the geometry of the sample.

## Data availability

The data that support the findings of this study are available from the corresponding authors upon reasonable request.

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

## Acknowledgements

M.I.E. is thankful to the Max Planck community for the support, and Prof. Dr. U. Pöschl for the constant encouragement. The authors thank Prof. Dr. M. Tkacz and Dr. M. A. Kuzovnikov for the synthesis of $LaH_3$ sample, Dr. G. Gu and Prof. A. Kaminski for providing single-crystal samples of $Bi_2Sr_2CaCu_2O_8$, Prof. P. C. Canfield, Mr. X. Xu and Prof. R. A. Ribeiro for providing single-crystal samples of $MgB_2$, and Dr. D. A. Knyazev for assistance in the preparatory work with SQUID. X-ray diffraction were performed at GeoSoilEnviro CARS (The University of Chicago, Sector 13), Advanced Photon Source (APS), Argonne National Laboratory and DESY (Hamburg, Germany), a member of the Helmholtz Association HGF. GeoSoilEnviro CARS is supported by the National Science Foundation-Earth Sciences (EAR-1634415) and Department of Energy-GeoSciences (DE-FG02-94ER14466). This research used resources of the Advanced Photon Source, a U.S. Department of Energy (DOE) Office of Science User Facility operated for the DOE Office of Science by Argonne National Laboratory under Contract No. DE-AC02-06CH11357. Parts of this research were carried out at PETRA-III using P02.2. Beamtime allocated for proposal I-20200636. Work at the Ames Laboratory (S.L.B.) was supported by the U.S. Department of Energy, Office of Science, Basic Energy Sciences, Materials Sciences and Engineering Division under Contract No. DE-AC02-07CH11358. The National High Magnetic Field Laboratory is supported by the National Science Foundation through NSF/DMR-1644779, the State of Florida, and the U.S. Department of Energy.

## Author contributions

V.S.M. and M.I.E. supervised the work. M.I.E. designed the miniature DAC. V.S.M. designed and performed the experiment. V.B.P., S.C., H.P.L. and R.J.H. assisted with the powder X-ray diffraction measurements. V.S.M., S.L.B. and F.F.B. processed and analyzed the data. V.S.M. and M.I.E. wrote the manuscript along with S.L.B. and F.F.B.

## Funding

## Competing interests

The authors declare no competing interests.
