## [Peer Review File · Nature Communications]

Reviewers' Comments:

Reviewer #1:

Remarks to the Author:

Claims of superconductivity in hydrogen-rich compounds have been based essentially on electrical transport measurements. A convincing and crucial test of superconductivity though is the observation of the Meissner effect and magnetic properties are important characteristics of a superconductor. Previous attempts to characterize the magnetic properties of H₃S and LaH₁₀ were not entirely convincing. In the present work, Minkov et al present more accurate SQUID measurements based on an ingenious approach to measure the reference magnetic signal of the whole DAC sample assembly before the synthesis of the superconducting compound. Also, test magnetic measurements on known superconductors (MgB₂ and Bi₂Sr₂CaCu₂O₈) are presented. The present work is a real advance for demonstrating the magnetic properties of these two iconic H-rich superconductors, H₃S and LaH₁₀. The paper is well-written, the experimental description is detailed enough and the analysis of the magnetization data correct. I recommend publication in Nature Communications.

The following few comments should be addressed first.

Comments :

- 1) The raw voltage curves of figure 2a & b should be discussed more. Looking at ref. 43, a dome shape should be more expected. What are the contributions modifying so much the raw signal?
- 2) The discussion on the integrity of the LaH₁₀ sample to explain no observation of superconductivity after the first heating of LaH₁₀ is not entirely convincing. Could it be that the stoichiometry is not correct? It has been reported that the stoichiometry of LaH_x can vary between various synthesis. The stoichiometry could be estimated from the XRD volume of LaH₁₀. Has it been checked from XRD data?
- 3) The magnetic measurements of the test materials should be better presented and discussed because that shows the feasibility of the SQUID measurements. Only the volume of the MgB₂ single crystal is comparable with the one of LaH₁₀ and H₃S. Also, I believe there has been an inversion between fig S12 d and fig S12c because the signal/noise ratio is much worse for measurements in the plastic tube than in the DAC. In that case, the measurements of the magnetic properties of MgB₂ with similar volume as the ones of the two hydrides measured has a very odd behavior. The other two test samples, MgB₂ powder and Bi₂Sr₂CaCu₂O₈ have volumes an order of magnitude larger. Measurements are as expected (there might also be an inversion between fig12S e & f). In its present form, the discussion of these test measurements is casting some doubts on the magnetization data obtained for the two hydrides.

Reviewer #2:

Remarks to the Author:

This paper reports an innovative approach to accurately measure the magnetic properties of the high-pressure superconductors H₃S and LaH₁₀. This is a remarkable achievement as high quality quantitative studies of the superconducting parameters of the H₃S and LaH₁₀ and by extension, other hydrogen-rich superconductors, have been limited so far. Demonstrating such measurements opens the door for further work on other newly discovered superconductors. I think the authors' demagnetization factor correction and the test of their setup on MgB₂ and Bi₂Sr₂CaCu₂O₈ are commendable. In my view, this paper deserves publication in Nature Communications after the authors have considered the comments below.

This manuscript reports the coherence length $\xi(0\text{ K}) \sim 1.8\text{ nm}$ in Im-3m-H₃S and $\sim 1.5\text{ nm}$ in Fm-3m-LaH₁₀. Those values are small and comparable to those of (unconventional) high-T_c cuprates. Typically, the coherence length values for BCS (phonon-mediated) superconductors are typically hundreds of nm (10.1038/s41598-021-91163-w, 10.1016/S1369-7021(08)70175-5). On the other hand, many papers including the authors' report and this manuscript, suggest LaH₁₀ and S₃H are conventional phonon-mediated superconductors. Could the short inferred coherence length be due to the short mean free path of the conduction carriers due to many defects or domain boundaries generated by a considerable strain under non-hydrostatic pressures exceeding 100 GPa? Is it possible that if the same measurements were performed on homogeneous samples under

hydrostatic conditions the results may be different?

The authors describe the superconducting transition signal as "subtle" in the field-cooling (FC) M (magnetization) vs T curve. But, at least for me, the transition in the FC curve is not visible (Fig. 2c-f). Is "subtle" really an accurate description?

In the supplemental material, for the sample thickness estimation, the authors describe that they have considered the increase of a sample volume due to the cupping deformation of anvils at high pressure. However, in the main text, the authors say that they have considered the volume expansion of the sample by hydrogenation. These descriptions seem to be in conflict. Moreover, even if the cupping deformation occurs, the distance between two diamond anvil tips decreases. As the sample thickness is a key to discuss the demagnetization factor in this manuscript, it is important that the authors clarify the discussion and make it consistent between the main text and the SI.

The reported values of H_{c1} are high; in most superconductors H_{c1} is in the order of a few hundreds of Gauss (MgB2 has a lower critical field approaching ~ 300 mT in the zero temp limit). But these hydrides are probably unique. Can the authors provide some discussion on this point?

Can they provide an estimate for the superconducting volume fraction they extract from these measurements? It is tricky since they do not know the exact size of the sample which is embedded under high pressure but if they can provide an estimate, it would be good.

The authors claim that their study places the superconducting hydride as a moderate type II superconductor. Can they explain what the implication of this conclusion for the nature of the superconducting state of hydrides would be.

I think they need to clarify the difference between the lower critical field H_{c1} and the penetration field H_p in Fig. 3 and on page 11. To me it was not clear. They have considered the demagnetization factor in reporting the values for H_{c1} , but I do not think demagnetization correction was performed when extracting the H_p values. The authors should clarify the discussion of these points.

Additional points

The meaning of PD, PH and PS in Fig. 1 should be explained.

line 190: $2.8 \cdot 10^6 \text{ A cm}^{-2} \rightarrow 2.8 \times 10^6 \text{ A cm}^{-2}$

line 218: I think 'The low values of κ indicate,,,' should be

'The low values of ξ indicate,,,'

How was equation (4) derived?

Is it possible to add a table summarizing all the superconducting parameters (such as H_{c1} , K, LL, etc) they estimated in their studies? This helps readers to see the results immediately.

In Fig S4, the curves for the Ampere's law are missing in the plot.

Reviewer #3:
None

RESPONSE TO REVIEWERS' COMMENTS

We are thankful to Reviewers for their valuable comments and suggestions, due to which the Manuscript has been significantly improved for readers. We accordingly modified the main text, figures and supplementary materials, paying special attention to those parts, which are written unclearly or discussed insufficiently. We used options of "Track Changes" and "Text Highlight" in the modified doc. files of the Revised Manuscript. Below we provide our responses to the Reviewers' comments.

Reviewer #1 (Remarks to the Author):

Claims of superconductivity in hydrogen-rich compounds have been based essentially on electrical transport measurements. A convincing and crucial test of superconductivity though is the observation of the Meissner effect and magnetic properties are important characteristics of a superconductor. Previous attempts to characterize the magnetic properties of H₃S and LaH₁₀ were not entirely convincing. In the present work, Minkov et al present more accurate SQUID measurements based on an ingenious approach to measure the reference magnetic signal of the whole DAC sample assembly before the synthesis of the superconducting compound. Also, test magnetic measurements on known superconductors (MgB₂ and Bi₂Sr₂CaCu₂O₈) are presented. The present work is a real advance for demonstrating the magnetic properties of these two iconic H-rich superconductors, H₃S and LaH₁₀. The paper is well-written, the experimental description is detailed enough and the analysis of the magnetization data correct. I recommend publication in Nature Communications. The following few comments should be addressed first.

Comments

1. The raw voltage curves of figure 2a &b should be discussed more. Looking at ref. 43, a dome shape should be more expected. What are the contributions modifying so much the raw signal?

The difference in the background signal of miniature DACs between our measurements and those reported in Ref 43 stems from different design of these cells. In Ref 43 the authors used the symmetric design of DAC and had the symmetric background signal, the center of which was presumed to correspond to the sample position. Our DAC is not symmetric with the diamond anvils, rhenium gasket and sample close to the end of the DAC, not at the center of the DAC. Therefore, the resulting voltage profile is also not symmetric and shifted towards the massive part of the DAC – the fixing nut and piston (see the figure below). That is also a reason why we use small steel piece to reliably find the sample position.

We modified supplementary figure S3 (added a photo of the miniature DAC scaled to the length of DC scans) and extended the discussions for figure 2a and 2b in the main text.

2. The discussion on the integrity of the LaH10 sample to explain no observation of superconductivity after the first heating of LaH10 is not entirely convincing. Could it be that the stoichiometry is not correct? It has been reported that the stoichiometry of LaHx can vary between various synthesis. The stoichiometry could be estimated from the XRD volume of LaH10. Has it been checked from XRD data?

The stoichiometry of the LaHx products can be estimated from the X-ray diffraction data using the reference data of La or LaH3 at *the same pressure values*.

Fm-3m-LaH10 has the highest T_c in the La-H system reaching ~ 253 K at 170 GPa (there is claims of higher T_c s though). Lanthanum hydrides with lower hydrogen content exhibit lower T_c s and have different crystal lattices (Drozdov et al.), which drastically manifest in X-ray diffraction patterns.

We added Figure S8 in Supplementary Materials, which demonstrates almost identical X-ray diffraction powder patterns (almost the same lattice parameters) of the Fm-3m-LaH₁₀ phase collected from the same spot of the sample after the first heating (1000 K) and second laser heating (2000 K). Importantly, after the second laser heating the sample became shinier (see photos). Therefore, we presume that the integrity of the sample considerably improved after the second treatment at 2000 K.

3. The magnetic measurements of the test materials should be better presented and discussed because that shows the feasibility of the SQUID measurements. Only the volume of the MgB2 single crystal is comparable with the one of LaH10 and H3S. Also, I believe there has been an inversion between fig S12 d and fig S12c because the signal/noise ratio is much worse for measurements in the plastic tube than in the DAC. In that case, the measurements of the magnetic properties of MgB2 with similar volume as the ones of the two hydrides measured has a very odd behavior. The other two test samples, MgB2 powder and Bi2Sr2CaCu2O8 have volumes an order of magnitude larger. Measurements are as expected (there might also be an inversion between fig12S e & f). In its present form, the discussion of these test measurements is casting some doubts on the magnetization data obtained for the two hydrides.

The test magnetic measurements of the known ambient-pressure superconductors were performed to demonstrate that i) superconducting transitions and ii) a lower critical field H_{c1} can reliably be detected for $\sim 100\text{-}\mu\text{m}$ -diameter samples residing in our miniature DAC. It was important to confirm that the diamagnetic character of the superconducting transitions in MgB2 and Bi2Sr2CaCu2O8 and observed T_{cs} are in good agreement between two different measurements using a plastic tube and the miniature DAC. The measured value of H_{c1} for the test single crystal of Bi2Sr2CaCu2O8 also agrees well with the literature data.

We did not try to find the critical size of a test sample, the superconducting transition of which can be detected in our SQUID. However, the limits can be estimated from the measured signal using appropriate demagnetization correction.

In contrast to the Im-3m-H3S and Fm-3m-LaH10 phases, the test superconductors are anisotropic with layered crystal lattices and, therefore, the magnetic response value strongly depends on the orientation of a sample in an applied magnetic field. The single crystal MgB2 was randomly oriented in either plastic tube and DAC. The thin-plate shaped single crystal of Bi2Sr2CaCu2O8 was placed between two anvils in DAC perpendicular to an applied magnetic field yielding the maximum demagnetization correction and maximum value of the measured signal. The same crystal was placed onto the semispherical bottom of the plastic capsule and fixed by oil droplet. In later the thin-plate shaped sample was not perpendicular to a magnetic field what resulted in a decrease of the measured signal.

We added the necessary comments into

	Supplementary text, which describes the test measurements.
--	--

Reviewer #2 (Remarks to the Author):

This paper reports an innovative approach to accurately measure the magnetic properties of the high-pressure superconductors H3S and LaH10. This is a remarkable achievement as high quality quantitative studies of the superconducting parameters of the H3S and LaH10 and by extension, other hydrogen-rich superconductors, have been limited so far. Demonstrating such measurements opens the door for further work on other newly discovered superconductors. I think the authors' demagnetization factor correction and the test of their setup on MgB2 and Bi2Sr2CaCu2O8 are commendable. In my view, this paper deserves publication in Nature Communications after the authors have considered the comments below.

1. This manuscript reports the coherence length $\xi(0\text{ K}) \sim 1.8\text{ nm}$ in Im-3m-H3S and $\sim 1.5\text{ nm}$ in Fm-3m-LaH10. Those values are small and comparable to those of (unconventional) high-T_c cuprates. Typically, the coherence length values for BCS (phonon-mediated) superconductors are typically hundreds of nm (10.1038/s41598-021-91163-w, 10.1016/S1369-7021(08)70175-5). On the other hand, many papers including the authors' report and this manuscript, suggest LaH10 and S3H are conventional phonon-mediated superconductors. Could the short inferred coherence length be due to the short mean free path of the conduction carriers due to many defects or domain boundaries generated by a considerable strain under non-hydrostatic pressures exceeding 100 GPa? Is it possible that if the same measurements were performed on homogeneous samples under hydrostatic conditions the results may be different?	The coherence length $\xi(0\text{ K})$ values for Im-3m-H3S and Fm-3m-LaH10 were evaluated in references 21, 23 using experimental values of H_{c2} and Ginzburg – Landau approach. For clean type-II superconductors in a weak coupling limit $\xi_{\text{BCS}}(0\text{ K}) = \hbar v_F / \pi \Delta$, since Δ is proportional to T_c, one would expect (since v_F is not changing significantly) for hydrides the values of $\xi(0\text{ K})$ lower than for other known BCS superconductors that have much lower T_c. In the dirty case the “effective coherence length” is $1/\xi_{\text{eff}} = 1/\xi_{\text{BCS}} + 1/l$ where l is a mean free path, so the coherence length will be even smaller. So indeed, high T_c values and possible small mean free path will contribute to small $\xi(0\text{ K})$ values.
2. The authors describe the superconducting transition signal as "subtle" in the field-cooling (FC) M (magnetization) vs T curve. But, at least for me, the transition in the FC curve is not visible (Fig. 2c-f). Is "subtle" really an accurate description?	We agree that the present description is not perfect and do corresponding corrections. The Meissner effect (magnetic field expulsion in the FC measurements) is subtle in both LaH10 and H3S at low magnetic fields 2 mT (not illustrated in main Figure 2, but summarized in Figure S1 and S2 in Supplementary Materials). The smoothed curves of the FC data have the superconducting transition with very small value of ΔM (see Figure below). At higher magnetic fields (4 mT and 10 mT) the transition is less pronounced or undetectable in the FC data. We would like to emphasize that specific features of the Meissner effect in HTSC and its strong suppression by the external field are

associated with the strong vortex pinning, which prevents the vortices inside the sample from shifting towards the sample surface even after crossing the $H_{c1}(T)$ line. The very low fields are favorable for the 100% Meissner effect, because only in this case the $H_{c1}(T)$ line is crossed in the vicinity of T_c where the critical current $j_c(T)$ is small and the pinning is weak. Currently we are studying the pinning effects in hydrogen-rich superconducting materials. Our preliminary results support the presence of very strong pinning in these materials (The study will be published soon in a separate article).

We added the smoothed FC curves at 2 mT in the Figure S1 and S2 in Supplementary Materials and changed the main text accordingly.

3. In the supplemental material, for the sample thickness estimation, the authors describe that they have considered the increase of a sample volume due to the cupping deformation of anvils at high pressure. However, in the main text, the authors say that they have considered the volume expansion of the sample by hydrogenation. These descriptions seem to be in conflict. Moreover, even if the cupping deformation occurs, the distance between two diamond anvil tips decreases. As the sample thickness is a key to discuss the demagnetization factor in this manuscript, it is important that the authors clarify the discussion and make it consistent between the main text and the SI.

We agree that the volume of the sample does not increase with pressure. It decreases due to the extrusion and pressure-induced compression.

In our estimations of the upper limit of a sample thickness we assume *i*) a complete conversion of precursors (with the minimum required amount of NH_3BH_3) into the final products H_3S and LaH_{10} and *ii*) the deformation of diamond anvils at high pressure. In order to estimate a sample chamber volume between two anvils, we used literature data on equilibrium thickness of metal gaskets and cupping effect of diamond anvils. We meant that this volume is larger in the case of cupping in comparison with the volume between two ideally flat anvils (see figure below).

	4. The reported values of H_{c1} are high; in most superconductors H_{c1} is in the order of a few hundreds of Gauss (MgB2 has a lower critical field approaching ~ 300 mT in the zero temp limit). But these hydrides are probably unique. Can the authors provide some discussion on this point?	The Reviewer has a good point and we have been thinking about that since we evaluated H_{c1} values for H3S and LaH10. From the definition, high values of H_{c1} (or low values of λ_L) are stemming from the high density of states in the superconducting state. Unfortunately, we just have no reliable evaluations of the superconducting parameters such as the superconducting gap 2Δ and density of states $g(\epsilon_F)$. Current literature provides very broad range for these values, which can hardly be used for a discussion here. We are still trying to design new experiments that can help us to probe these parameters directly.
5. Can they provide an estimate for the superconducting volume fraction they extract from these measurements? It is tricky since they do not know the exact size of the sample which is embedded under high pressure but if they can provide an estimate, it would be good.	Unfortunately, we are not able to estimate the superconducting volume fraction. As we pointed the values of demagnetization factor N determined from the measured ΔM are about 2 times smaller than those calculated from the estimated ideal geometry of samples. Formally, it means that the volume fraction is about 50%. But the real shape of the superconducting samples can differ from the ideal shape because we ignore variation of a thickness in samples and imperfections of sample integrity, which are, conversely, already included in the experimental value of ΔM. A magnetic field penetrates into a superconductor at these imperfections decreasing an effective demagnetizing factor N. Thus, the real value of superconducting volume fraction should be higher than 50%.
6. The authors claim that their study places the superconducting hydride as a _moderate_ type II superconductor. Can they explain what the implication of this conclusion for the nature of the superconducting state of hydrides would be.	Likely a moderate κ may result in weaker vortex fluctuations. We added the discussions of the Ginzburg-Levanyuk number G_i, which quantifies the scale of fluctuations responsible for vortex melting and vortex creep in a superconductor. $G_i(0 K) \sim 9 \times 10^{-7}$ for H₃S and $\sim 6 \times 10^{-6}$ for LaH₁₀, which is substantially smaller than what is reported for cuprate and pnictide high-temperature superconductors and comparable to that of Nb₃Sn. Despite high T_c both hydrides display a

	moderate κ which results in weaker vortex fluctuations and explains why the reported vortex liquid region remains narrow even at high magnetic fields.																								
7. I think they need to clarify the difference between the lower critical field H_{c1} and the penetration field H_p in Fig. 3 and on page 11. To me it was not clear. They have considered the demagnetization factor in reporting the values for H_{c1}, but I do not think demagnetization correction was performed when extracting the H_p values. The authors should clarify the discussion of these points.	We modified the main text as follows: “The value of H_p, at which the applied magnetic field starts to penetrate into the sample, was determined from the onset of the deviation of $M(H)$ from the linear dependence (see Figure 3).” – i.e. H_p is the measured value of H_{c1} in our thin disk-shaped samples. The raw $M(H)$ data, which were used to estimate H_p are shown in Figure 3a and 3b. The $H_p(T)$ dependence is shown in Figure 3c and 3d. In order to get the geometry-independent values of H_{c1} for a thin-disk shaped sample we need to apply the demagnetization correction. In the main text we discuss the values of $H_{c1}(0\text{ K})$ using different estimations of N.																								
8. Additional points: a) The meaning of PD, PH and PS in Fig. 1 should be explained. b) line 190: $2.8 \times 10^6 \text{ A cm}^{-2} \rightarrow 2.8 \times 10^6 \text{ A cm}^{-2}$; c) line 218: I think 'The low values of κ indicate,,,' should be 'The low values of ξ indicate,,,' d) How was equation (4) derived? e) Is it possible to add a table summarizing all the superconducting parameters (such as H_{c1}, K, LL, etc) they estimated in their studies? This helps readers to see the results immediately. f) In Fig S4, the curves for the Ampere’s law are missing in the plot.	a) Done. b) Done. c) the original text is correct. d) The equation was taken from Charles P. Poole, Jr., Horacio A. Farach, Richard J. Creswick, Superconductivity, Academic Press, San Diego, 1995 (equation 9.13b on page 270). e) Done. f) The Bean’s model originates from the Ampere’s law. In Figure S4 both models yield the same values of critical current densities. For the sake of simplicity, we show only one set of data in the revised Figure.   <caption>Estimated data points from the graph</caption>   Temperature (K) j_c ($\times 10^6 \text{ A cm}^{-2}$) for LaH_{10} j_c ($\times 10^6 \text{ A cm}^{-2}$) for H_3S     80 7.0 7.0   100 7.2 7.2   140 5.2 5.2   160 3.2 3.0   180 2.2 1.8   200 1.2 0.0   240 0.0 0.0   	Temperature (K)	j_c ($\times 10^6 \text{ A cm}^{-2}$) for LaH_{10}	j_c ($\times 10^6 \text{ A cm}^{-2}$) for H_3S	80	7.0	7.0	100	7.2	7.2	140	5.2	5.2	160	3.2	3.0	180	2.2	1.8	200	1.2	0.0	240	0.0	0.0
Temperature (K)	j_c ($\times 10^6 \text{ A cm}^{-2}$) for LaH_{10}	j_c ($\times 10^6 \text{ A cm}^{-2}$) for H_3S																							
80	7.0	7.0																							
100	7.2	7.2																							
140	5.2	5.2																							
160	3.2	3.0																							
180	2.2	1.8																							
200	1.2	0.0																							
240	0.0	0.0																							

Reviewers' Comments:

Reviewer #1:

Remarks to the Author:

My concerns have been satisfactorily addressed and my comments taken into account in this revised version.

I recommend publication.

Reviewer #2:

Remarks to the Author:

Referee's second report.

After reviewing the authors' response and revised manuscript, I feel the manuscript has been substantially improved and I recommend publication with no further changes required.

I do have one suggestion for the authors regarding Question 2 from Reviewer #2.

Looking at the smoothed Moment vs. T curves indicated in the rebuttal letter, I think it is likely that the authors have observed the Meissner effect in FC curves at 2 mT. However, the superconducting transition signal is small, within the error bars.

I would like to suggest showing the Fig. S1 middle-left panel and S3 middle-left panel in the main text to help readers understand and find the subtle signals in FC curves at 2 mT. Also, is it possible to compare the data at 2 mT with that of 4 mT in the same graphs? It would help readers understand the authors' claim of weak flux exclusion or its absence.

RESPONSE TO REVIEWERS' COMMENTS

We are thankful to Reviewers for their valuable comments and suggestions, due to which the Manuscript has been significantly improved. We accordingly modified the main text, figures and supplementary materials, paying special attention to those parts, which are written unclearly or discussed insufficiently. We used options of "Track Changes" and "Text Highlight" in the modified doc. files of the Revised Manuscript. Below we provide our responses to the Reviewers' comments.

Reviewer #1 (Remarks to the Author):

My concerns have been satisfactorily addressed and my comments taken into account in this revised version. I recommend publication.

Reviewer #2 (Remarks to the Author):

After reviewing the authors' response and revised manuscript, I feel the manuscript has been substantially improved and I recommend publication with no further changes required.

I do have one suggestion for the authors regarding Question 2 from Reviewer #2. Looking at the smoothed Moment vs. T curves indicated in the rebuttal letter, I think it is likely that the authors have observed the Meissner effect in FC curves at 2 mT. However, the superconducting transition signal is small, within the error bars.

I would like to suggest showing the Fig. S1 middle-left panel and S3 middle-left panel in the main text to help readers understand and find the subtle signals in FC curves at 2 mT. Also, is it possible to compare the data at 2 mT with that of 4 mT in the same graphs? It would help readers understand the authors' claim of weak flux exclusion or its absence.

Response:

We have modified Figure 2 in the main text and added additional panels, which show FC portions of $M(T)$ measurements for H3S and LaH10 samples.